# Sustained Disease Control in DME Patients upon Treatment Cessation with Brolucizumab

**DOI:** 10.3390/jcm13061534

**Published:** 2024-03-07

**Authors:** Justus G. Garweg, Sonja Steinhauer

**Affiliations:** 1Swiss Eye Institute and Clinic for Vitreoretinal Diseases, Berner Augenklinik, 3007 Bern, Switzerland; sonja.steinhauer@augenklinik-bern.ch; 2Department of Ophthalmology, Bern University Hospital, 3010 Bern, Switzerland

**Keywords:** diabetic macular edema, brolucizumab, treatment cessation, intravitreal therapy, anti-VEGF agents, case series, KITE&KESTREL studies

## Abstract

**Background:** Treatment cessation due to a dry retina has not been systematically addressed in diabetic macular edema (DME). In three out of four patients receiving 6 mg of brolucizumab in the KITE study, treatment was terminated after the study ended. **Methods:** The KITE study was a double-masked, multicenter, active-controlled, randomized trial (NCT 03481660) in DME patients. Per protocol, patients received five loading injections of Brolucizumab at 6-week intervals, with the option to adjust to 8 weeks in case of disease activity or to extend in the second year to a maximum of 16 weeks in the absence of retinal fluid. **Results:** After two years, one patient required eight weekly injections, while three patients reached a maximal treatment interval of 16 weeks. The severity of diabetic retinopathy improved in all patients with no dye leakage according to fluorescein angiography (FA) and no retinal fluid according to OCT in three patients. Treatment was paused in these three patients for >36 months, while the fourth patient required continuous treatment at 5-week intervals after switching to other licensed anti-VEGF agents. **Conclusions:** The adoption of treatment according to individual needs, including considering treatment cessation, may contribute to improved treatment adherence in many patients and be more frequently possible than expected.

## 1. Introduction

Intravitreal anti-vascular endothelial growth factor (VEGF) therapy in patients with diabetic macular edema (DME) is associated with meaningful improvements in vision and vision-related quality of life, which are more pronounced for near-body activities than for distant activities [1]. Nevertheless, clinical experience with typically multimorbid diabetic patients with DME proves their high treatment demand and low adherence. In real life, more appointments and treatments are missed in these patients than in patients with neovascular age-related macular degeneration (nAMD) [2], and their comorbidities are less well controlled than in other patients [3,4], resulting in significantly higher health care costs [5] and an increased risk of severe vision loss [6]. In contrast with those used for nAMD treatment, none of the existing anti-VEGF treatment protocols have been shown to be superior to the other protocols (fixed dosing compared with pro re nata (PRN) and treat-and-extend (T&E)) in terms of functional outcomes, although the treatment burden associated with these protocols is lowest with the T&E regimen [7].

Although clinical practice guidelines for DME generally recommend anti-VEGF agents as first-line therapy, they differ in their criteria for treatment initiation, such as for baseline visual acuity, location of DME, and central subfield thickness (CST); for retreatment; and for the use of therapeutic alternatives, i.e., intravitreal corticosteroids and focal photocoagulation [8,9,10]. This may explain the differences between randomized clinical trial outcomes and unfiltered real-word evidence, most frequently, but not exclusively, explained by limited compliance and treatment adherence, as outlined above [11]. In the absence of robust evidence, the majority of guidelines on how to improve real-world outcomes are grounded in expert opinion [12].

In nAMD, unsatisfactory long-term outcomes linked to undertreatment and limited treatment adherence [13] have triggered multiple studies aiming to optimize treatment strategies [14,15]. These studies, in summary, demonstrate that a loading phase is beneficial for early control of disease activity, followed by individualized proactive T&E therapy. These outcomes are equivalent to those of fixed treatment after two or more years, but with a reduced treatment burden and improved patient adherence, and are superior to those of partial nasal hernia syndrome (PRN) or as needed therapy in terms of functional outcomes [15,16,17,18]. Several retrospective cohort studies have assessed treatment cessation after reaching stability, i.e., the absence of retinal fluid over prolonged periods in nAMD patients [19,20,21]. There is generally broad agreement within the community that treatment interruption or cessation may be an interesting option for nAMD patients, although recurrences must be expected [21]. However, this option has not been included in the current treatment guidelines [22].

In DME patients, treatment cessation due to stability after 3 or more years of consequent treatment has been reported in several studies [8,9,23], but has never been systematically addressed. In the case of treatment interruption, on the other hand, nonadherence has frequently been assumed [11]. The heterogeneity of the diabetic population regarding the duration and control of diabetes, insulin dependence, and underlying comorbidities is strongly associated with the variable duration and morphological presentation of DME at baseline, which may be reflected in a highly variable short-term treatment response and less predictable long-term treatment demand [23]. Not surprisingly, few studies have reported treatment cessation in patients with DME receiving anti-VEGF therapy [4]. Here, we present a series of four patients with DME treated with brolucizumab for more than two years in the KITE study [24], three of whom were able to pause treatment for 3 years after the end of the study. Generally, more than 50% of patients with DME might qualify for long-term treatment cessation. The drying potential and the time until reaching a dry and stable retinal situation allowing treatment cessation might differ between marketed anti-VEGF drugs.

## 2. Materials and Methods

The KITE study was a double-masked, 100-week, multicenter, active-controlled, randomized trial (NCT 03481660) in patients with DME. Patients in the brolucizumab arm received five loading doses of brolucizumab (6 mg) every 6 weeks (q6w) followed by q12w dosing, with the option of adjusting to q8w at predefined disease activity assessment visits. Based on the disease stability assessment, treatment intervals could be extended by 4 weeks at week 72 [24]. Four of our patients were randomized into this group and treated per the protocol, while the fifth received aflibercept 2 mg. The last treatment interval under brolucizumab was blinded, and the last potential intravitreal bevacizumab injection visit was at week 96. In this single-center observational study, we report the evolution of the disease and treatment demand under brolucizumab after the end-of-study close-up visit at week 100 after baseline over the following 36 or more months. A fifth patient was included in the KITE study and randomized to the aflibercept arm receiving 5 monthly intravitreal injections of aflibercept 2 mg, thereafter bimonthly until end of the study. Based on minimal residual fluid, treatment was interrupted after week 100 for 5 months before he newly experienced recurrent DME, requiring treatment re-uptake. Five years after treatment initiation, aflibercept treatment was paused for a second time.

At the close-out visit, the full work-up included visual acuity assessment, intraocular pressure, clinical examination, OCT (6 × 6 mm^2^, 49 B-scans; HRT2, Heidelberg Engineering, Heidelberg, Germany), and wide-field imaging including angiography (California, Optos Inc., Dunfermline, Scotland). Except for angiography, all assessments were repeated bimonthly while wide-field angiography was repeated three years after the close-out visit.

## 3. Results

Four patients were randomized to the 6 mg of brolucizumab arm and received five loading injections per protocol at six-week intervals; thereafter, the treatment interval was extended to 12 weeks in the first year and to 16 weeks in the second year. If the blinded investigator reported disease activity at predefined disease activity assessments at weeks 32 and 36 and every 12 weeks thereafter, the treatment interval had to be shortened to 8 weeks and could not be extended until week 72, which was required by one of the four eyes. On the other hand, at this visit, no disease activity was reported, and a four-week extension from 8 to 12 or from 12 to 16 weeks was possible, which reached three out of our four patients. The fourth patient remained on an 8-week interval until the end of the study. Close-out clinical and fluorescein angiographic (FA) images as well as ocular coherence tomography (OCT) findings after 2 years (week 100) indicated an improvement in diabetic retinopathy severity in all instances (Figure 1, Figure 2, Figure 3 and Figure 4). While there was no macular or peripheral retinal disease activity, no major ischemia or dye leakage was found in three patients; this was also observed via OCT and FA in the fourth patient. An epiretinal membrane without traction was present at baseline in patients 1 and 3, but did not progress.

Three of the four patients had preexisting major cardiovascular disease, which included coronary heart disease (CHD) and intermittent atrial fibrillation (IF); treatment with 20 mg/d rivaroxaban (patient 1); treatment after myocardial infarction and coronary surgery; treatment after bilateral subtotal arteria carotis interna stenosis; treatment requiring endarterectomy (patient 2); treatment after myocardial infarction; and treatment after cerebrovascular insult (patient 3). No major vascular events were reported for the fourth patient (Table 1, demographic data). Her visual acuity and central retinal thickness responded well to the first injection in all instances, and diabetic retinopathy severity (DRS) improved from a baseline mild to moderate nonproliferative diabetic retinopathy (NPDR) in all cases, to no to mild NPDR in the first 2 years under brolucizumab treatment (Figure 1, Figure 2, Figure 3 and Figure 4, Table 2).

Based on the functionally excellent recovery, reaching a full visual acuity after visual gains of 7–17 letters (1.5 to 3.5 lines of vision) until the end of the study, treatment cessation was requested by three of the four patients (Figure 1, Figure 2 and Figure 3), while the treatment demand for the fourth patient increased from every 8 weeks under brolucizumab to every 4–5 weeks after the necessary switch to approved anti-VEGF drugs on the market (ranibizumab and aflibercept) to stabilize his visual function after the end of the study without allowing for the extension of the treatment interval over time (Figure 4). As the affected eye was still phakic, he decided against the use of corticosteroid implants.

## 4. Discussion

Treatment can be interrupted in a significant portion of eyes with DME after three to five years; this has been reported in several trials, such as RISE and RIDE [25], DRCR.net protocol I [26,27], protocol T [28], and VISTA and Vivid [29,30]; however, treatment cessation has never been systematically addressed in clinical treatment strategies. Possibly linked to the stronger drying potential of the new generation of anti-VEGF drugs, this may already be possible after two years if complete control of disease activity (i.e., a dry retina) is achieved, which may be linked to the greater drying potential of the new generation of anti-VEGF drugs, [31,32,33] as in this series, which was achieved with brolucizumab [34]. However, whether a higher drying potential will achieve stability and whether treatment cessation occurs earlier and/or more frequently with longer-acting drugs have yet to be demonstrated.

Since the introduction of anti-VEGF agents, clinically relevant endpoints of randomized clinical trials (RCTs) for DME have extensively been discussed. Until recently, visual acuity and central macular thickness were set as reproducible and robust endpoints in virtually all prospective comparative anti-VEGF studies by the national health authorities [35]. However, the clinical study outcomes have regularly been found to be superior to real life, where case selection criteria of RCTs cannot be applied [36,37,38,39]. Moreover, visual gains and change in CRT are comparable between the marketed drugs and are thus not able to differentiate between the different drugs on the market [40]. Given the broad availability of high-resolution OCT, different structural and functional biomarkers have been discussed [35], but none of them have achieved general acceptance [41]. In recent years, the presence of retinal fluid and retinal atrophy [32] have been used in RCTs for nAMD, but not for DME, though they would be excellent to compare the different drugs. Other biomarkers, such as the time to dryness and mean maximal treatment interval after 2 years may be of more interest in the clinical routine to describe the therapeutic potential of newer anti-VEGF agents. The time to achieve disease stability as a basis to consider treatment cessation has, although clinically meaningful, not been used in clinical trials, possibly because the typical duration of RCTs of two years may be too short to establish this bio-marker. With the more recent longer-acting drugs such as brolucizumab, faricimab, and aflibercept 8 mg, this might hopefully change based on recent RCT findings and triggered by our observations.

The advantages of using a T&E protocol to minimize the treatment burden for patients with DME on an individual basis have been well established [7], but obviously, 30% of patients extending to ≥12 weeks will interrupt treatment and be twice as likely to discontinue treatment than those with a ≤8-week treatment interval [11]. According to several large clinical trials, including the KITE and KESTREL studies, patients with DME are on average 10 years younger than patients with nAMD [24,42,43]. Multiple comorbidities are associated with a lower patient compliance and known to contribute to a limited treatment adherence in DME patients [5]. This could be improved by a partnership between patients and physicians in the decision process [44] and by ca lose cooperation among all relevant medical disciplines in diabetes care [45]. In clinical practice, the treatment demand for DME decreases markedly with increasing duration of anti-VEGF therapy [10,46], with 25% (ranibizumab) to 32% (aflibercept) of patients not requiring further anti-VEGF injection 3–5 years after it reaches stability [28,46]. Given that 3 out of our 4 patients did not require further injections after 2 years of treatment with brolucizumab for more than 36 months, it seems surprising that treatment cessation has never been systematically addressed. Based on our observations, two questions should be systematically addressed: (A) What are the possible criteria for treatment cessation, and how many patients qualify for treatment cessation after 2, 3, and 5 years of treatment based on these criteria? (B) Is the chance for and time to treatment cessation linked to the severity of the underlying disease and the preexisting vascular damage or to the anti-VEGF drug under which a dry retina has been removed?

Many guidelines have defined the roles of anti-VEGF and corticosteroid treatment in DME, including strategies to cope with cases insufficiently responsive to anti-VEGF agents [8,9,12,47,48]. The option of treatment cessation in eyes achieving a dry retina under stable ≥12-weekly intervals using anti-VEGF therapy has never systematically been considered [11]. Several reasons may have contributed to these findings: (i) Most of the guidelines, including the Euretina guidelines, were established many years ago when only ranibizumab and aflibercept were approved as anti-VEGF agents for the treatment of DME and when the likelihood of achieving a dry retina was considered low [9]. (ii) Since the advent of anti-VEGF therapy, DME has evolved from an irreversible and destructive outer retinal process caused by classical ETDRS thermal photocoagulation [49,50,51] to a more diffuse type of central visual handicap [52,53]. In contrast to advanced nAMD, central scotomata resulting from undertreatment are less of a concern [54]. (iii) DME may be accompanied by severe, but partially reversible vision loss. However, even eyes with clinically significant DME for ≥6 months may experience remarkable visual gains if treated adequately [55,56], which is rare in nAMD [57]. (iv) Compliance with scheduled visits and treatment adherence are less predictable in DME than in AMD patients [2]. Given that noncompliance is expected in more than 40% of patients with DME [42,43], the risk of a delayed diagnosis of DME recurrence due to noncompliance with scheduled visits is increased, arguing against treatment cessation; moreover, foggy but not so much distorted vision and a slowly progressive vision loss differentiate DME from nAMD [52,54], resulting in an increased risk of recurrent DME remaining undiagnosed until significant vision loss has been encountered [48]. Finally, the therapeutic response of eyes with DME may vary significantly between individuals, which renders predicting the control intervals and treatment demand difficult for individual patient-based decisions [58].

## 5. Conclusions

Individualization of therapy and the option to achieve disease stability while allowing for treatment cessation seem realistic for a majority of patients with this chronic disease. Although the duration of treatment interruption cannot currently be predicted, 3 of 4 own patients receiving brolucizumab reached this point after 2 years of treatment. Moreover, treatment has been interrupted for 3 years under regular clinical control. We hope that this case series will encourage opting for treatment cessation in cases with sufficiently controlled diabetic retinal disease. It may be considered as an additional endpoint in future prospective trials in DME. The potential of single anti-VEGF agents to reach this point has yet to be determined.

## Figures and Tables

**Figure 1 jcm-13-01534-f001:**
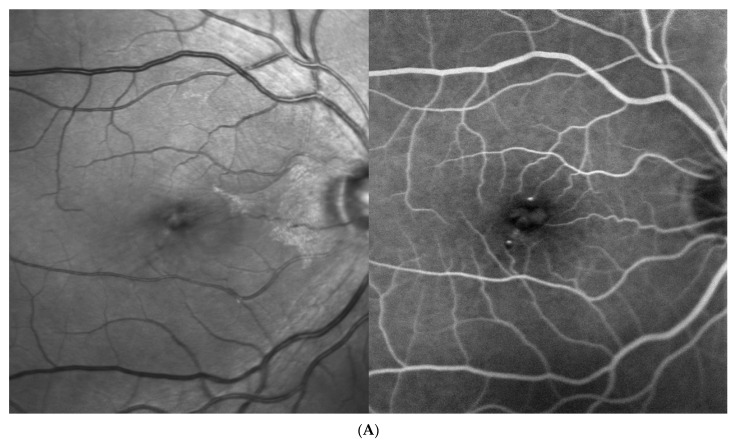
Patient 1, m, at diagnosis of diabetic macular edema 74 years old, with oral well-controlled diabetes mellitus type 2 for more than 20 years. (**A**) Redfree and late fluorescein angiographic images at treatment initiation. (**B**) Late fluorescein angiographic and OCT images after 2 years (left) and OCT at brolucizumab treatment initiation (baseline) and after 2 and 5 years (right).

**Figure 2 jcm-13-01534-f002:**
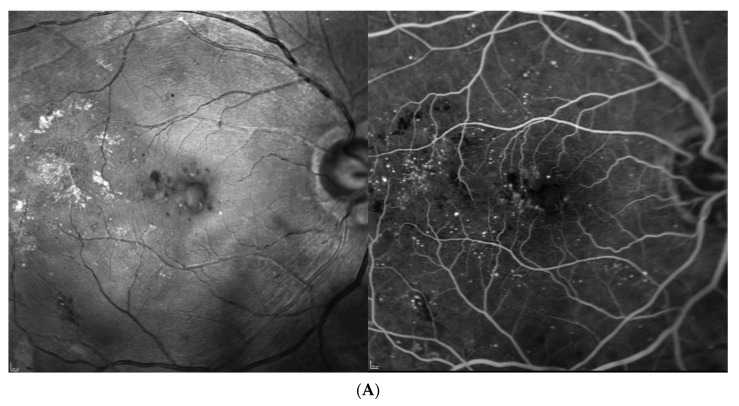
Patient 2, m, upon diagnosis of diabetic macular edema 78 years old, moderately controlled diabetes mellitus type 2 since 20 years, insulin dependent. (**A**) Redfree and late fluorescein angiographic images upon treatment initiation. (**B**) Redfree and late fluorescein angiographic images after 2 years. (**C**) OCT images at initiation of brolucizumab treatment (baseline) and after 2 and 5 years.

**Figure 3 jcm-13-01534-f003:**
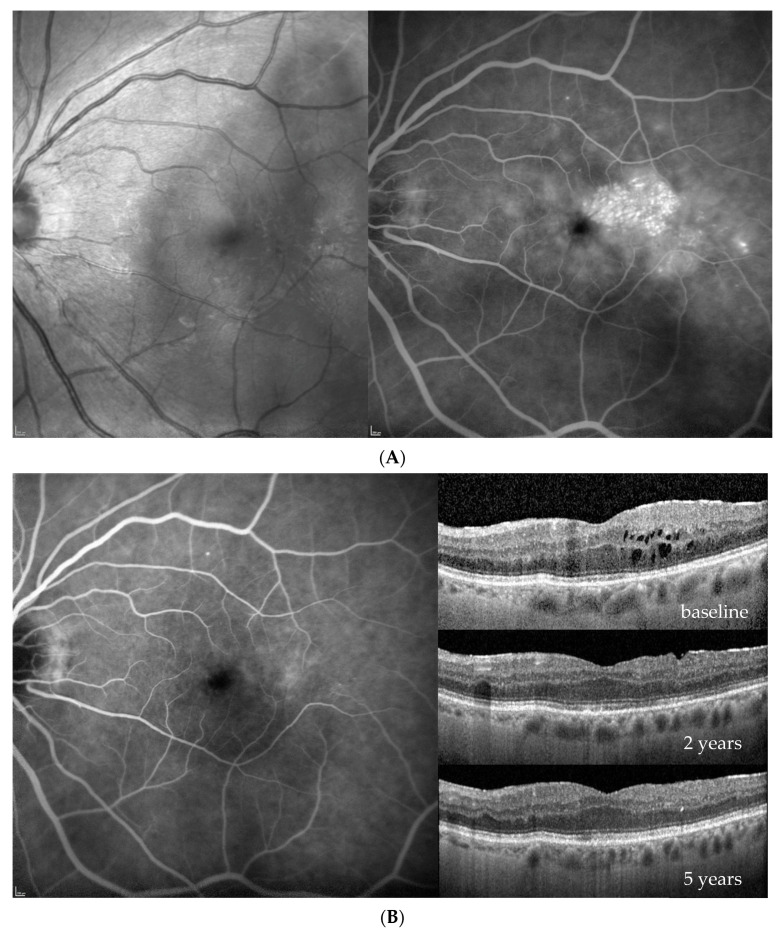
Patient 3, f, upon diagnosis of diabetic macular edema 73 years old, moderately controlled diabetes mellitus type 2 since 34 years, insulin dependent. (**A**) Redfree and late fluorescein angiographic images at treatment initiation. (**B**) late fluorescein angiographic images (left) and OCT images upon initiation of brolucizumab treatment (baseline) and after 2 and 5 years (right).

**Figure 4 jcm-13-01534-f004:**
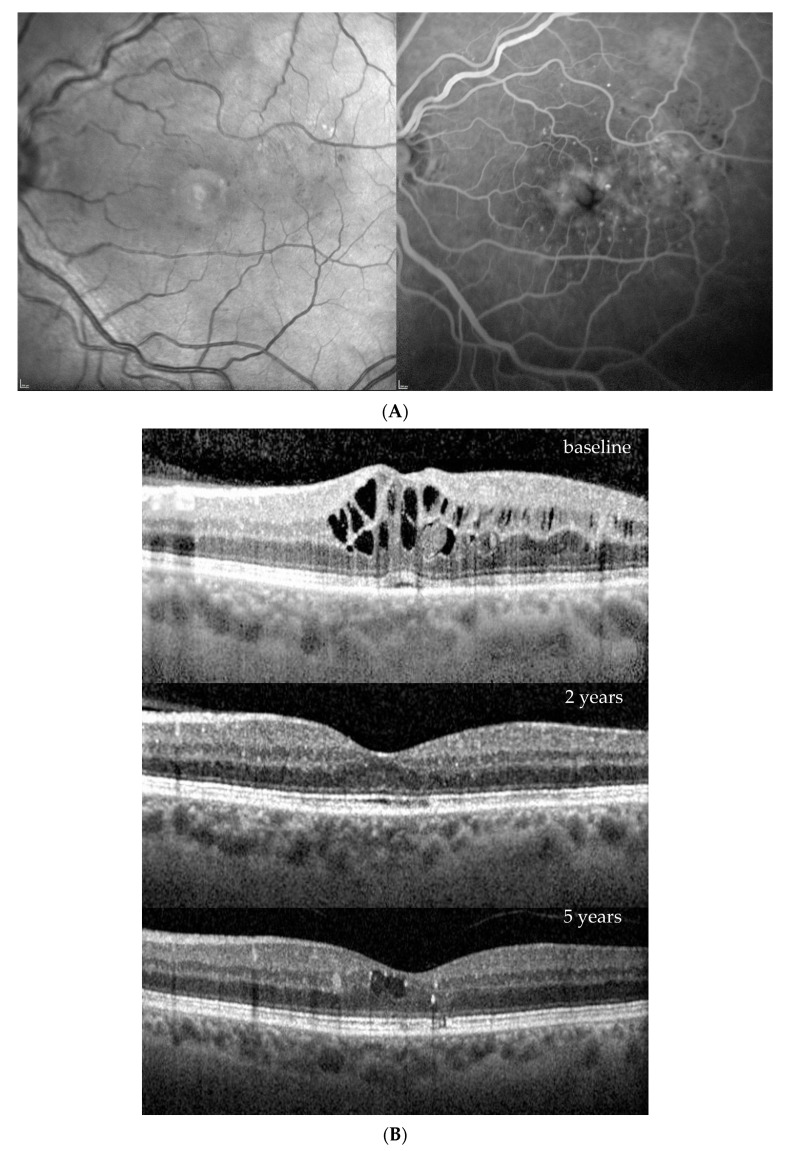
Patient 4, m, upon diagnosis of diabetic macular edema 66 years old, diabetes mellitus type 2 for 20 years, insulin dependent, except for controlled arterial hypertension, no comorbidity. (**A**) Redfree and late fluorescein angiographic images at treatment initiation. (**B**) OCT images upon initiation of brolucizumab treatment and after 2 and 5 years (4 weeks after the last intravitreal aflibercept injection).

**Table 1 jcm-13-01534-t001:** Baseline demographic data of the four patients.

Baseline (at DME Treatment Initiation)		Patient 1	Patient 2	Patient 3	Patient 4
Age	years	74	78	73	66
Gender	m/f	m	m	f	m
Body Mass Index	kg/m^2^	29.6	29.3	35.9	24.2
Diabetes type		2	2	2	2
Diabetes duration	years	20	13	34	20
Hb1aC	%	6.2	9	8.3	9.8
Insulin dependence		no	yes	yes	yes
Hypertension		yes	yes	yes	yes
Dyslipidemia		yes	no	no	no
Sleep apnea		yes	yes	yes	no
Smoking state		unknown	unknown	former smoker	former smoker
Regular sporting activity	hours per day	0	1.5	0	3
Major cardiovascular events *	number	1	2	2	0
Visual impairment due to DME in the partner eye		no	no	yes	no

* For details, see the text.

**Table 2 jcm-13-01534-t002:** Ocular findings of the four patients.

Affected Eye		R	R	L	L
BCVA	ETDRS score	77	73	69	73
	Snellen decimal	0.8	0.63	0.5	0.63
CST	µm	409	466	410	470
NPDR severity		mild	moderate	moderate	mild
Epiretinal membrane		yes	no	yes	no
Presence of vitreomacular traction		no	no	no	no
24 months					
BCVA	ETDRS score	90	81	86	80
	Snellen decimal	1.25	0.8	1.0	0.8
CST	µm	330	274	337	591
NPDR severity		background	mild	mild	mild
36 months					
BCVA	Snellen decimal	1	1	1	0.8
CST	µm	345	302	361	365
48 months					
BCVA	Snellen decimal	1.25	1.0	1.0	1.0
CST	µm	339	266	344	293
60 months					
BCVA	Snellen decimal	1.0	1.25	1.0	0.63
CST	µm	342	263	340	513

NPDR, nonproliferative diabetic retinopathy; BCVA, best-corrected visual acuity; CST, central retinal subfield thickness.

## Data Availability

All data presented in this study have been displayed in the text, tables and figures. The raw data supporting the conclusions of this article may also be made available by the authors on request.

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
