# Peer review of "Sustained Disease Control in DME Patients upon Treatment Cessation with Brolucizumab"

_jcm, 2024, doi:10.3390/jcm13061534_

Round 1

Reviewer 1 Report

Comments and Suggestions for Authors

While there is abundant literature on therapeutic strategies for DME (Diabetic Macular Edema), the issue of treatment cessation due to a dry retina has not been systematically addressed. This paper introduces several cases of DME patients treated with Brolucizumab through case reports, proposing the option of treatment cessation in eyes achieving a dry retina under stable 12-weekly intervals using anti-VEGF therapy. This has some scientific value, but there are some major concerns and suggestions:

1. The reason why there are few systematic studies on treatment cessation is primarily due to the interference of many factors in determining the timing of DME (Diabetic Macular Edema) treatment cessation. These factors include systemic factors in diabetic patients, the severity of diabetic retinopathy, the specific type and initiation time and intervals of anti-VEGF treatment used, compliance, medication changes, and the presence of other treatments, among others. Therefore, the treatment cessation scheme proposed by the authors, based merely on 3 out of 4 cases, is difficult to be convincing. However, as case reports, they do have a certain inspirational effect, and the authors also mention in the conclusion that "The time to treatment cessation and/or the number of eyes reaching this point may be linked to the drying potential of the anti-VEGF drug," although this is not comprehensive. I suggest that the authors more thoroughly discuss in the discussion section the existing research on the endpoints of DME treatment, including any conflicts, consensus, influencing factors, and the limitations of this case series.

2. In the introduction and discussion sections, there is an excessive discussion on the therapeutic effects of anti-VEGF for AMD (Age-related Macular Degeneration), which is unnecessary.

3. The description in the methods section is insufficient, including the equipment used for FA (Fluorescein Angiography) and OCT (Optical Coherence Tomography), scanning range, etc., as well as the frequency of follow-up in the 3 years after the study cessation, what other treatments were involved, and so on.

4. My understanding is that the KITE study is a multicenter clinical trial, and the several cases in the authors' case series are part of this multicenter clinical trial, involving patients from the institution where the authors are based. It remains unclear whether other centers involved in the study have similar cases. Additionally, clinical trials often feature multiple groups; it is not specified whether patients included in other treatment groups have or have not reached this point after 2-year treatment.

5. Please label composite images with numbers in the Figures section and provide descriptions in the figure legends.

6. In Table 1, please describe the type of diabetes of the patients.

7. In the conclusion section, it is stated that "Our findings indicate that compliance and adherence are limited in patients with DME, especially when baseline visual acuity is low and baseline age is advanced." However, I believe the findings of this case series do not support this conclusion, which seems to be derived from a literature review. Please reorganize the Conclusion section to better analyze the value and limitations of this case series, as well as future research directions.

Author Response

Reviewer 1:

While there is abundant literature on therapeutic strategies for DME (Diabetic Macular Edema), the issue of treatment cessation due to a dry retina has not been systematically addressed. This paper introduces several cases of DME patients treated with Brolucizumab through case reports, proposing the option of treatment cessation in eyes achieving a dry retina under stable ≥12-weekly intervals using anti-VEGF therapy. This has some scientific value, but there are some major concerns and suggestions:

Response:

Thanks for a principally favourable vote.

Reviewer 1:

  1. The reason why there are few systematic studies on treatment cessation is primarily due to the interference of many factors in determining the timing of DME (Diabetic Macular Edema) treatment cessation. These factors include systemic factors in diabetic patients, the severity of diabetic retinopathy, the specific type and initiation time and intervals of anti-VEGF treatment used, compliance, medication changes, and the presence of other treatments, among others. Therefore, the treatment cessation scheme proposed by the authors, based merely on 3 out of 4 cases, is difficult to be convincing. However, as case reports, they do have a certain inspirational effect, and the authors also mention in the conclusion that "The time to treatment cessation and/or the number of eyes reaching this point may be linked to the drying potential of the anti-VEGF drug," although this is not comprehensive.

I suggest that the authors more thoroughly discuss in the discussion section the existing research on the endpoints of DME treatment, including any conflicts, consensus, influencing factors, and the limitations of this case series.

Response:

We fully agree with the reviewer that the multiple confounders and the co-morbidities limit predicting treatment outcomes as outlined in the introduction. We do, on the other hand, not provide any cessation scheme, while we clearly state that we aim to induce a critical thinking of the option of treatment cessation.

Here, we wished to present a concept including treatment cessation, while this case series does not justify an elaboration of frequently used other study endpoints. Given that most studies have follow up times of 2, maximally 3 years and have as yet not included treatment cessation as an end point, we feel that the portion of eyes reaching this endpoint possibly differs between the single drugs. The wording of the conclusion has therefore remained vague: “The potential of the single anti-VEGF agents to reach this point has as yet to be determined.”

The limitation of this study is clearly that it is a case series, which has clearly been stated. Additionally, we added the following text to the last paragraph: “We hope that this case series will encourage to opt for treatment cessation in cases with sufficiently controlled diabetic retinal disease and may be used as endpoint in future prospective trials.”

Reviewer 1:

  1. In the introduction and discussion sections, there is an excessive discussion on the therapeutic effects of anti-VEGF for AMD (Age-related Macular Degeneration), which is unnecessary.

Response:

This information in the introduction aimed to provide some background on the state of the art and has been shortened as requested.

Reviewer 1:

  1. The description in the methods section is insufficient, including the equipment used for FA (Fluorescein Angiography) and OCT (Optical Coherence Tomography), scanning range, etc., as well as the frequency of follow-up in the 3 years after the study cessation, what other treatments were involved, and so on.

Response:

Thanks for pointing on this. The following text was added: “At the close-out visit, the full work-up included besides visual acuity assessment, intraocular pressure, clinical examination, OCT (6x6mm2, 49 B-scans; HRT2, Heidel-berg Engineering, Heidelberg, Germany) and wide-field imaging including angiography (California, Optos Inc., Dunfermline, Scotland). Except angiography, all assessments were repeated bimonthly while wide-field angiography was repeated three years after the close-out visit.”

Reviewer 1:

  1. My understanding is that the KITE study is a multicenter clinical trial, and the several cases in the authors' case series are part of this multicenter clinical trial, involving patients from the institution where the authors are based. It remains unclear whether other centers involved in the study have similar cases. Additionally, clinical trials often feature multiple groups; it is not specified whether patients included in other treatment groups have or have not reached this point after 2-year treatment.

Response:

Indeed, this is a multicenter study, but we were due to data protection issues not able to contact other centers to share their follow up findings. Moreover, this was not a pre-planned post-study follow up. Therefore, the text was supplemented by: “In this single-center observational study”

Reviewer 1:

  1. Please label composite images with numbers in the Figures section and provide descriptions in the figure legends.

Response:

Done as requested

Reviewer 1:

  1. In Table 1, please describe the type of diabetes of the patients.

Response:

All patrients had type 2, now added to the table

Reviewer 1:

  1. In the conclusion section, it is stated that "Our findings indicate that compliance and adherence are limited in patients with DME, especially when baseline visual acuity is low and baseline age is advanced." However, I believe the findings of this case series do not support this conclusion, which seems to be derived from a literature review. Please reorganize the Conclusion section to better analyze the value and limitations of this case series, as well as future research directions.

Response:

We agree and have re-worded the conclusion to: “Individualization of therapy and the option to achieve disease stability while allowing treatment cessation seem realistic for a majority of patients with this chronic disease. Although the duration of treatment interruption cannot currently be predicted, 3 of 4 patients receiving brolucizumab reached this point after 2 years of treatment. Moreover, treatment has meanwhile been interrupted for 3 years under regular clinical controls. We hope that this case series will encourage to opt for treatment cessation in cases with sufficiently controlled diabetic retinal disease and may be used as endpoint in future prospective trials in DME. The potential of the single anti-VEGF agents to reach this point has as yet to be determined.”

Reviewer 2 Report

Comments and Suggestions for Authors

This is an interesting case series supplementary to the KITE study, where four DME patients undergoing brolucizumab injection were randomised into this group. The authors are particularly interested with this group because DME patients have a greater tendency to be lost in follow-up, and therefore risk having poorer treatment outcomes. While examining this group of patients, the authors were able to achieve improvement by individualising their treatment plans. They concluded that the individualised treatment plans help to improve compliance. The manuscript is generally well written, but the authors should focus their conclusion based on the results from this present study. For example, the first two lines in the conclusion section is citing a different study, and this should belong in the discussion section instead. To improve readability, lines 90 to 96 should be under the methods section rather than the results section. Although the results are tabulated nicely in a table, the authors should mention key information in the main text so that it is easier for readers to follow. Some of the description in the main text is also quite vague. For example, the authors mentioned about good treatment response but did not clearly state how much the improvement is in terms of visual acuity and retinal thickness. These minor corrections should help to improve the readability of the manuscript. 

Author Response

Reviewer 2:

This is an interesting case series supplementary to the KITE study, where four DME patients undergoing brolucizumab injection were randomised into this group. The authors are particularly interested with this group because DME patients have a greater tendency to be lost in follow-up, and therefore risk having poorer treatment outcomes. While examining this group of patients, the authors were able to achieve improvement by individualising their treatment plans. They concluded that the individualised treatment plans help to improve compliance.

The manuscript is generally well written, but the authors should focus their conclusion based on the results from this present study. For example, the first two lines in the conclusion section is citing a different study, and this should belong in the discussion section instead.

Response:

We agree and have re-worded the conclusion to: “Individualization of therapy and the option to achieve disease stability while allowing treatment cessation seem realistic for a majority of patients with this chronic disease. Although the duration of treatment interruption cannot currently be predicted, 3 of 4 patients receiving brolucizumab reached this point after 2 years of treatment. Moreover, treatment has meanwhile been interrupted for 3 years under regular clinical controls. We hope that this case series will encourage to opt for treatment cessation in cases with sufficiently controlled diabetic retinal disease and may be used as endpoint in future prospective trials in DME. The potential of the single anti-VEGF agents to reach this point has as yet to be determined.”

Reviewer 2:

To improve readability, lines 90 to 96 should be under the methods section rather than the results section.

Response:

Thanks for this input. The reported treatment extension was based on week 72 findings, which is a result of importance for treatment extension and interruption. We therefore decided to leave the lines unchanged in Results.

Reviewer 2:

Although the results are tabulated nicely in a table, the authors should mention key information in the main text so that it is easier for readers to follow.

Response:

The following text is now found in results and summarizes the two tables: “Three of the four patients had preexisting major cardiovascular disease, which included coronary heart disease (CHD) and intermittent atrial fibrillation (IF); treatment with 20 mg/d rivaroxaban (patient 1); treatment after myocardial infarction and coronary surgery; treatment after bilateral subtotal arteria carotis interna steno-sis; treatment requiring endarterectomy (patient 2); treatment after myocardial infarction; and treatment after cerebrovascular insult (patient 3). No major vascular events were reported for the fourth patient (Table 1, demographic data). Her visual acuity and central retinal thickness responded well to the first injection in all in-stances, and diabetic retinopathy severity (DRS) improved from a baseline mild to moderate nonproliferative diabetic retinopathy (NPDR) in all cases to no to mild NPDR in the first 2 years under brolucizumab treatment (Figures 1-4, Table 2).”

Reviewer 2:

Some of the description in the main text is also quite vague. For example, the authors mentioned about good treatment response but did not clearly state how much the improvement is in terms of visual acuity and retinal thickness. These minor corrections should help to improve the readability of the manuscript.

Response:

Thanks for pointing on this. The text was now modified to: “Based on the functionally excellent recovery, reaching a full visual acuity after visual gains of 7-17 letters (1.5 to 3.5 lines of vision) until end of the study, treatment cessation was requested by three of the four patients (Figures 1-3), while the treatment demand for the fourth patient increased from every 8 weeks under brolucizumab to every 4-5 weeks after the necessary switch to approved anti-VEGF drugs on the market (ranibizumab and aflibercept) to stabilize his visual function after the end of the study without allowing extension of the treatment interval over time (Figure 4).”

Round 2

Reviewer 1 Report

Comments and Suggestions for Authors

Thank you for the response; most of my concerns have been addressed satisfactorily, and some issues raised in the comments have been resolved and enhanced in the text. However, "I suggest that the authors more thoroughly discuss in the discussion section the existing research on the endpoints of DME treatment, including any conflicts, consensus, influencing factors, and the limitations of this case series" was not well incorporated.

Author Response

jcm-2891894.R1

Reviewer 1:

Thank you for the response; most of my concerns have been addressed satisfactorily, and some issues raised in the comments have been resolved and enhanced in the text. However, "I suggest that the authors more thoroughly discuss in the discussion section the existing research on the endpoints of DME treatment, including any conflicts, consensus, influencing factors, and the limitations of this case series" was not well incorporated.

Response:

We are happy to be allowed to discuss clinical study end points as far as supported by clinical evidence which can be found in the discussion now. The new text reads: “Since the introduction of anti-VEGF agents, clinically relevant endpoints of randomized clinical trials (RCTs) for DME have extensively been discussed. Until recently, visual acuity and central macular thickness were set as reproducible and robust endpoints in virtually all prospective comparative anti-VEGF studies by the nation-al health authorities [35]. However, the clinical study outcomes have regularly been found superior to real life, where case selection criteria of RCTs cannot be applied [36-39]. Moreover, visual gains and change in CRT are comparable between the marketed drugs and thus not able to differentiate between the different drugs on the market [40]. Given the broad availability of high-resolution OCT, different structural and functional biomarkers have been discussed [35], but none of them has achieved general acceptance [41]. In recent years, the presence of retinal fluid and retinal atrophy [42] have been used in RCTs namely for nAMD and would be excel-lent to compare the different drugs. Other biomarkers, such as the time to dryness and mean maximal treatment interval after 2 years may be of more interest in the clinical routine and have been introduced to describe the therapeutic potential of newer anti-VEGF agents. The time to achieve disease stability as a basis to consider treatment cessation, has, though clinically meaningful, not been used in clinical tri-als, possibly because the typical duration of RCTs of two years may be too short to establish this bio-marker. With the more recent longer-acting drugs such as brolucizumab, faricimab, and aflibercept 8mg, this might possibly change based on recent RCT findings and triggered by our observations.”